# Development and Evolution of DNA-Dependent Protein Kinase Inhibitors toward Cancer Therapy

**DOI:** 10.3390/ijms23084264

**Published:** 2022-04-12

**Authors:** Yoshihisa Matsumoto

**Affiliations:** Laboratory for Zero-Carbon Energy, Institute of Innovative Research, Tokyo Institute of Technology, Tokyo 152-8550, Japan; yoshim@zc.iir.titech.ac.jp; Tel.: +81-3-5734-2273

**Keywords:** DNA double-strand break (DSB), non-homologous end joining (NHEJ), DNA-dependent protein kinase (DNA-PK), phosphatidylinositol 3-kinase, inhibitor, radiosensitization, chemosensitization

## Abstract

DNA double-strand break (DSB) is considered the most deleterious type of DNA damage, which is generated by ionizing radiation (IR) and a subset of anticancer drugs. DNA-dependent protein kinase (DNA-PK), which is composed of a DNA-PK catalytic subunit (DNA-PKcs) and Ku80-Ku70 heterodimer, acts as the molecular sensor for DSB and plays a pivotal role in DSB repair through non-homologous end joining (NHEJ). Cells deficient for DNA-PKcs show hypersensitivity to IR and several DNA-damaging agents. Cellular sensitivity to IR and DNA-damaging agents can be augmented by the inhibition of DNA-PK. A number of small molecules that inhibit DNA-PK have been developed. Here, the development and evolution of inhibitors targeting DNA-PK for cancer therapy is reviewed. Significant parts of the inhibitors were developed based on the structural similarity of DNA-PK to phosphatidylinositol 3-kinases (PI3Ks) and PI3K-related kinases (PIKKs), including Ataxia-telangiectasia mutated (ATM). Some of DNA-PK inhibitors, e.g., NU7026 and NU7441, have been used extensively in the studies for cellular function of DNA-PK. Recently developed inhibitors, e.g., M3814 and AZD7648, are in clinical trials and on the way to be utilized in cancer therapy in combination with radiotherapy and chemotherapy.

## 1. Introduction

Ionizing radiation (IR) is thought to exert a variety of biological effects through the induction of damages on DNA. One Gy of X-ray or γ-ray is estimated to induce approximately 500 thymine glycols, 150 DNA-protein crosslinks, 1000 single-strand breaks, and 40 double-strand breaks (DSBs) [1]. DSB is considered the most deleterious among the various types of DNA damage.

In eukaryotes, DSB is repaired mainly through homologous recombination (HR) and non-homologous end joining (NHEJ) [2]. There are two other pathways, i.e., alternative end joining (A-EJ) and single-strand annealing (SSA) [2]. A-EJ is also termed microhomology-mediated end joining (MMEJ) or DNA polymerase theta-mediated end joining (TMEJ). These four pathways are distinguished by their usage of sequence homology (Figure 1). HR reconstitutes the DNA sequence around DSB using a homologous or identical sequence as the template, which is usually longer than 100 base pairs (bp). On the other hand, NHEJ utilizes little or no sequence homology, i.e., 0–4 bp. A-EJ and SSA utilize sequence homology of 2–20 bp and more than 50 bp, respectively. HR, A-EJ and SSA are preceded by the end resection, which creates single-stranded DNA with 3′-overhang. The end resection proceeds in two stages, i.e., initial short-range resection (≈100 nucleotides (nt)) followed by long-range resection (several hundred or thousand nt). While A-EJ requires only short-range resection, HR and SSA require long-range resection.

In NHEJ, the DNA ends that are not compatible for ligation undergo end processing, which results in the deletion or insertion of nucleotides at the junction. In addition, joining of the ends in close vicinity may sometimes lead to ligation of incorrect pairs of DNA ends, resulting in chromosomal aberrations such as deletions, inversions, and translocations. Thus, NHEJ is considered more error-prone than HR. However, HR in vertebrates has a requirement for the sister chromatid and is restricted to late S and G2 phases. (Note: a very recent study demonstrated that DSBs at the centromere are repaired through HR even in the G1 phase [3].) The majority of cells are in G1 and G0 phases, in which cells rely on NHEJ to repair DSBs. In human cells, NHEJ accounts for approximately 80% of DSB repair even in the G2 phase [2]. Moreover, in most cases, the deletion or insertion of a small number of nucleotides can be tolerated, because only a small portion of the genome encodes proteins. A-EJ and SSA are thought to be more error-prone than NHEJ, because they are apt to occur between repetitive sequences, resulting in the loss of the sequence in between.

NHEJ is also implicated in the process of V(D)J recombination in vertebrate immune system [2]. Enormous diversity of immunoglobulins and T cell receptors are generated through the recombination of V (variable), D (diversity), and J (joining) segments, each of which can be selected from a number of segments. Recombination activating gene 1 and 2 (RAG1 and RAG2) induce a cleavage between the selected segments and the flanking recombination signal sequences. Then, the segments are joined through NHEJ. Thus, NHEJ is thought to be of prominent importance especially in vertebrates such as humans.

## 2. DNA-PK and Its Role NHEJ

DNA-dependent protein kinase (DNA-PK) is composed of a DNA-PK catalytic subunit (DNA-PKcs) and Ku heterodimer (hereafter denoted Ku), which consists of Ku80 (also termed Ku86) and Ku70 [4,5]. DNA-PK binds to and is activated by the end of a double-stranded DNA (dsDNA). Thus, DNA-PK acts as the sensor for the end of dsDNA, which appears when a DSB is generated.

DNA-PKcs is a huge protein consisting of 4128 amino acids (Figure 2A) [6]. DNA-PKcs is structurally related to Ataxia-telangiectasia mutated (ATM) and ATM- and Rad3-related (ATR) kinases, which are also implicated in DNA repair and DNA damage response [6,7,8,9]. ATM is recruited to DSB by the MRN complex consisting of Mre11, Rad50 and Nbs1, the last of which is responsible for Nijmegen breakage syndrome [10,11]. ATR is recruited to ssDNA through interaction with ATR-interacting protein (ATRIP) and Replication Protein A (RPA) [12]. These kinases show structural similarity to phosphatidylinositol 3-kinases (PI3Ks) and assemble the PIKK family. There are three additional PIKK members in humans, i.e., mammalian Target of rapamycin (mTOR, also termed FKBP12-rapamycin-associated protein, FRAP, and Rapamycin and FKBP12-target, RAFT) [13,14], Suppressor of morphological defects on genitalia-1 (SMG-1) [15,16], and Transformation/transcription domain-associated protein (TRRAP) [17]. Interestingly, TRRAP lacks kinase catalytic activity [17]. In addition to the kinase domain, these proteins share the FAT (FRAP, ATM and TRRAP), PRD (PIKK-regulatory domain) and FATC (FAT C-terminal) domains (Figure 2A). The primary function of mTOR is the regulation of cell growth and survival [13,14]. SMG-1 is essential for nonsense-mediated mRNA decay (NMD) [15,16]. PI3Ks mediates the signals from G-protein coupled receptors and receptor tyrosine kinases through the activation of AKT protein kinase (also known as protein kinase B, PKB) and mTOR [18,19,20]. This signaling pathway is called the PI3K/AKT/mTOR pathway. There are lines of evidence implicating these molecules in DNA damage response, although less directly than DNA-PKcs, ATM, and ATR [18,19,20]. SMG-1 was shown to activate the G1/S checkpoint through p53 upregulation and Cdc25A downregulation [21,22]. Recent studies showed that the human papilloma virus E6 protein and DNMT1 enhance radiosensitivity via the downregulation of SMG-1 [23,24]. The PI3K/AKT/mTOR pathway is upregulated in response to radiation and promotes cell survival [25]. AKT inhibits apoptosis through the downregulation of proapoptotic proteins, such as B cell lymphoma 2 associated agonist of cell death (BAD) [26,27] and upregulation of antiapoptotic proteins, such as human homolog of murine double minute 2 (HDM2), which promotes the degradation of p53 [28]. It is also reported that AKT augments NHEJ and HR [29,30]. Moreover, DNA-PK is shown to activate AKT in response to DNA damage directly or indirectly via Sty1/Spc1-interacting protein 1 (Sin1) [31,32]. The upregulation of the PI3K/AKT/mTOR pathway is frequently found in various types of cancer and is associated with resistance to radiotherapy and chemotherapy [18,19,20]. Thus, mTOR and PI3Ks are considered promising targets for radiosensitization and chemosensitization.

Ku80 and Ku70 consist of 732 and 609 amino acids, respectively (Figure 2B) [33,34]. Ku was initially identified as the antigen against autoantibody in a patient with an autoimmune disease, scleroderma-polymyositis overlap syndrome. Ku binds to the end of dsDNA without any particular preference in the nucleotide sequence [35]. DNA-PKcs is recruited to the end of dsDNA via interaction with the C-terminal region of Ku80 [36].

X-ray crystallography showed that Ku forms a ring-shaped structure that can encircle DNA, accounting for how Ku binds selectively to DNA ends [37]. Recent cryoelectron microscopy (cryo-EM) studies revealed a structure of DNA-PKcs complexed with Ku and DNA [38] (Figure 2C). DNA-PKcs is folded into a ring and a head. The ring includes the HEAT repeats and the interface with Ku, while the head includes FAT, kinase, PRD, and FAT-C domains [38]. DNA is inserted into the rings of Ku and DNA-PKcs [38]. The structural differences between inactive and active states of DNA-PKcs are also revealed. Most notably, PRD is closed in the inactive state and is assumed to clash with the substrate polypeptide. However, PRD becomes open in the active state, allowing the entry of a substrate polypeptide. Another cryo-EM study revealed the structure of DNA-PKcs in a complex with adenosine-5′-(γ-thio)-triphosphate (ATPγS), which is a non-hydrolyzable ATP analog [39] (Figure 2D). The adenine group is inserted into a hydrophobic pocket surrounded by Tyr3791, Trp3805 and Leu3806 [39]. Three phosphate groups and Mg^2+^ ions come in contact with Asn3926, Asn3927, Asp3941, Ser3731 and Lys3753 [39]. These multiple interactions are thought to stabilize the interaction between DNA-PKcs and ATP. While Lys3753, Tyr3791, Trp3805, Asn3927 and Asp3941 are fully conserved among PIKKs, Ser3731, Leu3806 and Asn3926 are divergent. The structures of DNA-PKcs with inhibitors were also elucidated (see below).

Ku and DNA-PKcs have been shown to be essential for NHEJ (Figure 2E and for details, refer to another review [40]). Initially, Ku80 was shown to correspond to X-ray repair cross-complementing group (XRCC) 5, which is deficient in a series of rodent cell lines exhibiting hypersensitivity to IR and defective V(D)J recombination [41,42]. Subsequently, DNA-PKcs was shown to correspond to XRCC7 and to be the responsible gene for murine severe combined immunodeficiency (*scid*) mutation [43,44,45]. Thereafter, a number of cells and animals deficient for DNA-PKcs, Ku80 or Ku70 were found or generated through gene targeting or genome editing [40]. In addition, six human individuals that harbor homozygous or compound heterozygous mutations in DNA-PKcs have been identified [40].

NHEJ proceeds in three stages (Figure 2E). In the recognition stage, Ku first binds to the end of DNA and then recruits DNA-PKcs. Paralog of XRCC4 and XLF (PAXX) stabilizes the binding of Ku to DNA and facilitates the subsequent assembly of the NHEJ factors [45,46,47,48]. When DNA ends are not compatible, they undergo the processing stage (for details, refer to review [2]). Artemis, in a complex with DNA-PKcs, exerts endonuclease activity on hairpin and overhang structures and 5′ to 3′ exonuclease activity on single-stranded DNA [49,50]. DNA polymerase μ (Polμ) and DNA polymerase λ (Polλ) fill in the gaps in DSBs. Polynucleotide kinase phosphatase (PNKP) adds a phosphate group at the 5′-end if absent and removes the phosphate group present at the 3′-end. Aprataxin (APTX) removes adenosinemonophosphate (AMP) from the abortive intermediates of ligation. Tyrosyl-DNA phosphodiesterase 1 (TDP1) and TDP2 remove the covalently bound proteins and phosphoglycolate groups from 3′-ends. In the ligation stage, DNA ligase IV (LIG4), which is associated with XRCC4, joins two DNA ends together [51,52,53]. XRCC4-like factor (XLF, also known as Cernunnos) forms filaments with XRCC4, which are suggested to align or bridge two DNA ends [54,55,56]. 

The kinase activity of DNA-PKcs is required for NHEJ because the catalytically inactive (kinase-dead) form of DNA-PKcs cannot rescue the radiosensitivity and V(D)J recombination defects of DNA-PKcs-deficient cells [57,58]. Although the precise roles of protein phosphorylation by DNA-PKcs remain elusive, DNA-PKcs is shown to phosphorylate NHEJ factors and other potentially NHEJ-related proteins (Table 1). The significance of phosphorylation of each substrate protein has been discussed elsewhere [59].

A number of small molecules that inhibit DNA-PK have been developed to date. These compounds have been powerful tools to delineate the function of DNA-PK. Furthermore, they are promising agents in cancer therapy, sensitizing cancer cells to radiotherapy and chemotherapy. Hereafter, the development of DNA-PK inhibitors and their potential in cancer therapy are reviewed.

## 3. Development and Evolution of DNA-PK Inhibitors–Pursuit for Potency and Selectivity

Since the discovery of the importance of DNA-PK in DSB repair through NHEJ, a number of small molecules inhibiting DNA-PK were developed from the 1990s to 2000s. Most of the inhibitors were developed on the basis of the structural similarity of DNA-PK to PI3K. In this phase, high potency, i.e., low 50% inhibiting concentration (IC_50_), and selectivity, i.e., high IC_50_ for other kinases especially PIKKs and PI3Ks were pursued. Some of the products, such as NU7026 and NU7441, were useful tools for the functional studies on DNA-PK.

### 3.1. OK-1035

The first reported inhibitor OK-1035, 3-cyano-5-(4-pyridyl)-6-hydrazonomethyl-2-pyridone (Figure 3) was found after screening over 10,000 natural and synthetic compounds [61]. The IC_50_ for DNA-PK activity in vitro was initially reported to be 8 μM [61] but was later reported to be 100 μM [62] (Table 2). OK-1035 retarded the DNA repair in cultured murine leukemia cells at 2 mM [63]. Although the IC_50_ of OK-1035 was at least 100-fold higher for other kinases such as Protein Kinase C, the effects of OK-1035 on PI3Ks and PIKKs were not tested. OK-1035 suppressed the accumulation of p53 and the induction of p21 in response to adriamycin treatment, suggesting that it might have inhibited ATM and/or ATR as well [64].

### 3.2. Wortmannin

The findings on the structural similarity of DNA-PK to PI3Ks and PIKKs paved a new avenue toward the development of DNA-PK inhibitors. Wortmannin, [(1R,3R,5S,9R,18S)-18-(methoxymethyl)-1,5-dimethyl-6,11,16-trioxo-13,17-dioxapentacyclo [10.6.1.02,10.05,9.015,19]nonadeca-2(10),12(19),14-trien-3-yl] acetate (Figure 4), which has been known as a PI3K inhibitor [65], was shown to inhibit DNA-PK [11]. Subsequently, ATM was also shown to be sensitive to wortmannin [66]. The IC_50_ of wortmannin for PI3K was 3.0 nM for PI3K [65], 16–120 nM for DNA-PK [67,84], and 100–150 nM for ATM [66,84] (Table 2). Wortmannin did not appreciably affect ATR [84]. Wortmannin was shown to covalently bind to DNA-PKcs [67]. A cryo-EM study, which was published this year, revealed the structure of DNA-PKcs with ATP-γS and four inhibitors, including wortmannin [39]. Wortmannin was shown to occupy the ATP binding site, forming a covalent bond to Lys3753 [39]. A number of studies published in the late 1990s to early 2000s showed that wortmannin augmented IR sensitivity and inhibited DSB repair at 20–50 μM [85,86]. Wortmannin showed radiosensitization on both DNA-PKcs-deficient and ATM-deficient cells [85]. Hence, the radiosensitizing effects of wortmannin are thought to be mediated through multiple mechanisms involving DNA-PK, ATM and PI3Ks.

### 3.3. LY294002-Derived Inhibitors

Another PI3K inhibitor LY294002, 2-(4-morpholinyl)-8-phenyl-4H-1-benzopyran-4-one (Figure 5A) [87], was also reported to inhibit DNA-PK [67]. IC_50_ of LY294002 was reported to be 6 μM for DNA-PK [84] and 1.4 μM for PI3K [56] (Table 2). Thus, LY294002 is not a selective inhibitor for DNA-PK, but it led to the discovery of more potent and selective inhibitors of DNA-PK, such as NU7026 and NU7441 (Figure 5). LY294002 was shown to enhance cellular radiosensitivity at 50 μM [85].

By screening a library of LY294002 derivatives, NU7026, 2-(morpholin-4-yl)-benzo[h]chromen4-one (Figure 5B), was found [69,88]. The IC_50_ of NU7026 for DNA-PK was 230 nM, which was much lower than that of ATM, ATR, mTOR and PI3K [69,88] (Table 2). The morpholine ring structure appeared essential for inhibitory activity. NU7026 at 10 μM sensitized cultured cells to radiation in a manner dependent on DNA-PK [69,88] (Table 3). NU7026 was also shown to potentiate the cytotoxicity of topoisomerase II poisons [89]. A preclinical pharmacokinetics study was conducted, showing rapid plasma clearance of NU7026 through metabolism [90]. Recent studies demonstrated that NU7026 administered intraperitoneally (i.p.) at 25–50 mg/kg could potentiate the tumor growth suppression via radiation and chemotherapeutic drugs salinomycin and TRAIL-inducing compound 10 (TIC10) in vivo, i.e., in xenograft in immunodeficient mice [91,92,93] (Table 3).

NU7441, 8-dibenzothiophen-4-yl-2-morpholin-4-yl-chromen-4-one (Figure 5C), showed more potent inhibition of DNA-PK than NU7026 [70,94]. The IC_50_ of NU7441 for DNA-PK was 14 nM [70,94] (Table 2). The latest structural study by cryo-EM showed the insertion of the chromen and morpholine groups into the deepest hydrophobic pocket of DNA-PKcs formed by Leu3751, Tyr3791, Ile3803, Leu3986 and Ile3940 and the insertion of the dibenzothiophene group into another hydrophobic pocket formed by Met3729, Pro3735 and Leu3751 [39]. These multiple interactions between NU7441 and DNA-PKcs would explain the higher affinity and selectivity of NU7441 than wortmannin for DNA-PKcs. To date, NU7441 has been most frequently used in functional studies of DNA-PK. NU7441 sensitized cultured cells to IR and etoposide in a manner dependent on DNA-PKcs at 0.5 μM [95] (Table 3). NU7441, 10–25 mg/kg, i.p., could potentiate tumor growth suppression by radiation and chemotherapeutic drugs in vivo [95,96] (Table 3).

KU-0060648, 2-(4-ethyl-piperazin-1-yl)-N-(4-(2-morpholino-4-oxo-4H-chromen-8-yl)-dibenzo[b,d]thiophen-1-yl)acetamide (Figure 5D), was developed by the modification of NU7441 to increase water solubility [68,97]. KU-0060648 exhibited an IC_50_ of 5 nM for DNA-PK, which is still lower than NU7441 but also inhibited PI3Ks at lower concentrations [68] (Table 2). Hence, KU-0060648 acts as a dual inhibitor for DNA-PK and PI3Ks. Growth inhibition was observed above 30 nM in cultured cancer cell lines and above 10 mg/kg in tumor xenografts [97,98,99] (Table 3). Sensitization to chemotherapeutic drugs was observed in similar dose ranges [97,98,99] (Table 3).

LTURM34, 8-(dibenzo[b,d]thiophen-4-yl)-2-morpholino-4H-1,3-benzoxazin-4-one (Figure 5E), in which the chromenone structure in NU7441 was replaced by benzoxazinone, was identified as a more selective inhibitor for DNA-PK [72]. While IC_50_ for DNA-PK was comparable to or higher than NU7441, IC_50_ for PI3Ks was more than two orders of magnitude higher [72] (Table 2). LTURM34 was shown to restore partial chemosensitivity to chemoresistant prostate cancer cells at 3 μM [100] (Table 3).

Recently, another NU7441-derivative NU5455, *N*-(6-(2-(8-oxa-3-azabicyclo [3.2.1]octan-3-yl)-4-oxo-4*H*-chromen-8-yl)dibenzo[b,d]thiophen-2-yl)-N-methyl-2-morpholinoacetamide (Figure 5F), was developed [73]. The IC_50_ of NU5455 for DNA-PK was 8.2 nM but was more than 30-fold higher for PI3Ks [73] (Table 2). In cellulo, NU5455 inhibited DNA-PKcs autophosphorylation with an IC_50_ of 168 nM and increased radiosensitivity and chemosensitivity at concentrations higher than 300 nM [73] (Table 3). Oral (p.o.) administration of NU5455 at 30–100 mg/kg potentiated tumor growth inhibition by radiation and chemotherapeutic agents in vivo (Table 3), notably without adverse effects in normal tissues [73]. 

It might also be noted that LY294002 was also used to derive ATM inhibitors, KU-55933, 2-morpholin-4-yl-6-thianthren-1-yl-pyran-4-one (Figure 5G) [101], and KU-60019, 2-((2R, 6S)-2,6-Dimethyl-morpholin-4-yl)-N-[5-(6-morpholin-4-yl-4-oxo-4H-pyran-2-yl)-9H-thioxanthen-2-yl]-acetamide (Figure 5H) [102].

### 3.4. Arylmorpholine-Based Inhibitors

IC60211, 2-hydroxy-4-morpholin-4-yl-benzaldehyde (Figure 6A), was found by screening the small molecular library and showed an IC_50_ of 400 nM for DNA-PK [74] (Table 2). It is noteworthy that this compound also includes a morpholine ring structure, which is similar to LY294002-derived inhibitors. Through the modification of IC60211, more potent and selective inhibitors for DNA-PK, including IC86621, 1-(2-hydroxy-4-morpholin-4-yl-phenyl)-ethanone (Figure 6B), and IC87361, 5-hydroxy-7-morpholino-2-phenyl-4H-chromen-4-one (Figure 6C), were obtained [74]. IC86621 and IC87361 exhibited radiosensitization and chemosensitization concomitantly with reduction in DSB repair ability [74] (Table 3). IC86621 administered subcutaneously (s.c.) at 400 mg/kg and IC87361 administered i.p. at 75 μg per mouse potentiated tumor growth inhibition in vivo [74,120] (Table 3).

Recently, SN38023, 5-((1-methyl-2-nitro-1H-imidazol-5-yl)methoxy)-7-morpholino-2-phenyl-4H-chromen-4-one (Figure 6D) was developed [121]. SN38023 itself showed less potent DNA-PK inhibition than IC87361 because of the presence of the nitroimidazole moiety, but it could be metabolized to IC87361 in hypoxic conditions [121]. Thus, SN38023 is expected to act as a prodrug for IC87361, which can target hypoxic tumor cells.

Through testing the arylmorpholine compound library for PI3Ks and PIKKs, AMA37, 1-(2-Hydroxy-4-morpholin-4-yl-phenyl)-phenyl-methanone (Figure 6E), was found as a selective inhibitor for DNA-PK [75] (Table 2). AMA37 showed radiosensitization at 20 μM [103] (Table 3).

### 3.5. Vanillin-Based Inhibitors

Vanillin, 4-hydroxy-3-methoxybenzoaldehyde (Figure 7A), is a natural product existing in plants such as *Vanilla planifolia*, which is mainly used in food industries as a flavoring agent. Vanillin was shown to inhibit DNA-PK at a high concentration, i.e., IC_50_ = 1500 μM [76] (Table 2). Vanillin also inhibited DNA end joining in cell-free extract and sensitized cells to cisplatin [76]. Screening the library of vanillin-related compounds led to the discovery of more potent inhibitors, such as 4,5-dimethoxy-2-nitrobenzaldehyde (DMNB) (Figure 7B) and 2-bromo-4,5-dimethoxybenzaldehyde (Figure 7C), with IC_50_ values of 15 μM and 30 μM, respectively [76] (Table 2). Although the continuous exposure of cells to 100 μM DMNB is toxic to the cells, one hour of treatment significantly increased cellular chemosensitivity [76] (Table 3). Interestingly, a vanillin derivative VND3207, 4-hydroxy-3,5-dimethoxybenzaldehyde (Figure 7D), exerted radioprotective rather than radiosensitizing effects, which was probably through radical scavenging activity and potentiation of DNA-PK activity [122,123].

### 3.6. SU11752

SU11752, 5-[[1,2-dihydro-2-oxo-5-[(phenylamino)sulfonyl]-3H-indol-3-ylidene]methyl]-2,4-diethyl-1H-pyrrole-3-propanoic acid (Figure 8), was identified by screening the three-substituted indoline-2-one library. The IC_50_ of SU11752 was comparable to wortmannin for DNA-PK (130 nM) but was higher for PI3K (1.1 μM) [77] (Table 2). Thus, SU11752 is considered a more selective inhibitor for DNA-PK than wortmannin. In cellulo, the inhibition of DSB repair was seen at 12–50 μM and radiosensitization was seen at 50 μM [77] (Table 3).

### 3.7. PI103

PI103, 2-(3-hydroxyphenyl)-4-morpholinopyrido [30,20:4,5]furo [3,2-d]pyrimidine (Figure 9A), was initially identified as a selective inhibitor for PI3Kα [124] but was subsequently found to inhibit mTOR [78] and DNA-PK as well [79,125]. The IC_50_ of PI103 for DNA-PK was 7.5 nM, which is comparable to that for PI3Kα and PI3Kβ (Table 2). PI103 enhanced the cellular radiosensitivity and chemosensitivity and retarded the DSB repair at 0.06–1 μM [104] (Table 3). Recently, a prodrug of PI103, i.e., RIDR-PI103, 2-amino-N-(5-amino-2-(3-(4-morpholinopyrido [3′,2′:4,5]furo [3,2-d]pyrimidin-2-yl)phenoxy)phenyl)acetamide (Figure 9B) was developed [126,127]. Reactive oxygen species stimulate the intramolecular circularization of this compound, resulting in the release of PI103 [126,127].

## 4. Development and Evolution of DNA-PK Inhibitors–Pursuit for Clinical Availability

The potent and selective inhibitors developed above may have been anticipated for applications in cancer therapy, but this was not feasible due to pharmacokinetics and toxicity. In the 2010s, additional inhibitors were developed and are now under clinical trials. Some of them are not selective inhibitors for DNA-PK and are even more inhibitory to mTOR and/or PI3Ks. There are also selective inhibitors for DNA-PK, which are expected to be used in combination with radiation and chemotherapeutic agents. In general, the inhibitors for clinical use show increased solubility in water for oral availability. They also tend to have a large structure and be inserted deeply into the ATP-binding pocket, being in contact as well with amino acids which are not conserved among PIKKs or not in contact with ATP (Figure 2D). Thus, increased contact will enhance the potency and/or selectivity of these inhibitors.

### 4.1. NVP-BEZ235 (Dactolicib)

NVP-BEZ235, 2-methyl-2-(4-(3-methyl-2-oxo-8-(quinolin-3-yl)-2,3-dihydro-1H-imidazo [4,5-c]quinolin-1-yl)phenyl)propanenitrile (Figure 10A), was initially identified as an orally available inhibitor for PI3K and mTOR [128], but it was found to inhibit DNA-PK, ATM and ATR as well [79,80] (Table 2). NVP-BEZ235 at 100 nM showed more potent radiosensitization and attenuation of DSB repair in comparison to NU7026 and KU55933 at 10 μM [80] (Table 3). Since NVP-BEZ235 could sensitize DNA-PKcs-deficient cells and ATM-deficient cells, radiosensitization might be due to the inhibition of DNA-PK and ATM [105]. NVP-BEZ235 administered p.o. at 50–75 mg/kg potentiated tumor growth inhibition by radiation in vivo [106] (Table 3). 

Phase 1 and 2 clinical trials are in progress, and the results of eight phase 1 studies and three phase 2 studies have been published to date. Most of these studies have attempted monotherapy, and inhibition of PI3K and/or mTOR rather than DNA-PK may be expected. To date, NVP-BEZ235 has not proven to be satisfactory in therapeutic efficacy and tolerability ([129,130,131] and others). 

ETP-46464, 2-methyl-2-(4-(2-oxo-9-(quinolin-3-yl)-2H-[1,3]oxazino [5,4-c]quinolin-1(4H)-yl)phenyl)propanenitrile (Figure 10B), which has a highly similar structure to NVP-BEZ235, was found as a selective inhibitor for ATR [131].

### 4.2. LY3023414 (Samotolisib)

LY3023414, 8-[5-(1-hydroxy-1-methylethyl)pyridin-3-yl]-1-[(2S)-2-methoxypropyl]-3-methyl-1,3-dihydro-2H-imidazo [4,5-c]quinolin-2-one (Figure 10C), was developed as a water-soluble and orally available inhibitor for PI3Ks [81]. LY3023414 is structurally similar to NVP-BEZ235. The administration of LY3023414 at 3–30 mg/kg p.o. showed growth inhibition and chemosensitization in vivo [81] (Table 3).

Phase 1 and 2 clinical trials are in progress, and the results of three monotherapeutic phase 1 studies [132,133,134] and one phase 2 study [135] have been reported to date. There is a report of a phase 1 study combining LY30234014 with Notch inhibitor crenigacestat (LY3039478). In these trials, the inhibition of PI3K and/or mTOR rather than DNA-PK may be expected. Three monotherapeutic phase 1 studies have shown tolerable safety properties with a recommended phase 2 dose (RP2D) of 200 mg administered twice daily (BID) [78,124,125]. In the phase 2 study, recruiting cancer patients with activating PI3K mutations showed only modest clinical activity [135]. The combination of crenigacestat and LY3023414 exhibited poor tolerance, which resulted in lowering the dose and reduced clinical activity [136]. 

### 4.3. CC-115

CC-115, 1-Ethyl-7-(2-methyl-6-(1H-1,2,4-triazol-3-yl)pyridin-3-yl)-3,4-dihydropyrazino [2,3-b]pyrazin-2(1H)-one (Figure 11), was found as a dual inhibitor for DNA-PK and mTOR, with IC_50_ of 13 nM and 21 nM, respectively [82] (Table 2). CC-115 inhibited in cellulo the NHEJ of plasmid reporters at 2–6 μM [137] and also sensitized cells to IR at 1 μM [107] (Table 3). It was shown to inhibit the growth of ATM-deficient cells, suggesting possible synthetic lethality by the simultaneous inactivation of DNA-PK and ATM [82]. The administration of CC-115 at 2–5 mg/kg p.o. showed growth inhibition in vivo [138].

Phase 1 and 2 clinical trials are in progress, and the results of two phase 1 studies have been published to date, which shows that the administration of 10 mg BID p.o. as monotherapy was tolerable and the clinical activity for advanced solid and hematopoietic malignancies was promising [139,140]. Further clinical studies on the combinatorial treatment with CC-115 and a DNA-damaging anticancer agent may be anticipated.

### 4.4. VX-984 (M9831)

VX-984, (S)-N-methyl-8-(1-((2’-methyl-[4,5’-bipyrimidin]-6-yl-4’,6’-d2)amino)propan-2-yl)quinoline-4-carboxamide (Figure 12), was found to be a potent and selective inhibitor for DNA-PK, although the IC_50_ of VX-984 for DNA-PK, other PIKKs, and PI3Ks in vitro is not described [108,141]. VX-984 was shown to sensitize cultured cells to IR with inhibition of DSB repair at 0.1–0.5 μM and to augment the tumor growth inhibition by radiation in xenograft at 50 mg/kg by p.o. administration in vivo [108,141] (Table 3). Phase 1 and 2 clinical trials of VX-984 monotherapy and combination with doxorubicin are currently in progress.

### 4.5. M3814 (Peposertib, Nedisertib)

M3814, (S)-[2-chloro-4-fluoro-5-(7-morpholinoquinazolin-4-yl)phenyl]-(6-methoxypyridazin-3-yl)methanol (Figure 13), was discovered through drug library screening at Merck KGaA [83,110]. The IC_50_ for DNA-PKcs was 0.6 nM and 20 nM in the presence of 10 μM or 1 mM ATP, respectively [110] (Table 2). After testing its effects on 284 lipid or protein kinases, only eight showed the IC_50_ values below 1 μM [108]. Thus, M3814 showed high selectivity to DNA-PK. M3814 inhibited DNA-PKcs autophosphorylation of Ser2056 in cellulo at 0.1–1 μM [110]. A recent structural analysis by cryo-EM indicated that morpholine and quinazoline groups fit well to the deepest hydrophobic pocket, similarly to NU7441 (see 3.3) [39]. In addition, its chloro-fluorobenzene ring interacts with Met3729, Ser3731, Pro3735, Leu3751 and Ile3940, and its pyridazine group fits the groove formed by Met3729, Trp3805, Thr 3811, Asn3926 and Met3929, which would stabilize the interaction [39].

M3814 enhanced cellular radiosensitivity in a manner dependent on DNA-PKcs at submicromolar concentration [110] (Table 2). M3814 also sensitized cells to chemotherapeutic agents, such as calicheamicin, microtubule polymerization inhibitor, i.e., paclitaxel, and topoisomerase II inhibitors, i.e., daunorubicin and etoposide [112] (Table 2). Whereas p53-proficient cells undergo cell cycle checkpoint, senescence or apoptosis upon treatment, p53-defcient cells undergo death by mitotic catastrophe [109,111]. Oral or intragastrical (i.g.) administration of M3814 at 5–100 mg/kg augmented tumor growth suppression by radiation and by chemotherapeutic agents, Milotarg (humanized anti-CD33 antibody conjugated to N-acetyl γ-calicheamicin), paclitaxel, etoposide and pegylated liposomal daunorubicin (PLD) in vivo [109,110,111,112,113,114] (Table 3). One of the latest studies showed that M3814 potentiated the in vivo tumor growth suppression by chemoradiotherapy using 5-fluorouracil [115] and radioimmunotherapy using bintrafusp α, which inhibit profibroblasitic TGFβ and immunosuppressive PD-L1 [116]. M3814 was also shown to bind to ATP-binding cassette transporter family G2 (ABCG2) and reverse drug resistance [142].

In a phase 1 study, M3814 was well tolerated with RP2D of 400 mg BID [143]. Concentration-dependent reduction in DNA-PKcs autophosphorylation in peripheral blood mononuclear cells was also observed [143]. Although M3814 monotherapy did not show partial response in this study [143], the combination with radiotherapy or chemotherapy is currently under phase 1 clinical trial, and the results are anticipated.

### 4.6. AZD7648

AZD7648, 7-methyl-2-[(7-methyl [1,2,4]triazolo [1,5-a]pyridin-6-yl)amino]-9-(tetrahydro-2H-pyran-4-yl)-7,9-dihydro-8H-purin-8-one (Figure 14), was identified by library screening followed by optimization in AstraZeneka [71,144]. In a panel of 397 kinases, only DNA-PK, PI3Kα, PI3Kβ and PI3Kδ showed >50% inhibition at 1 μM [71]. The IC_50_ for in vitro DNA-PK was 0.6 nM and that for in cellulo DNA-PKcs Ser2056 autophosphorylation was 91.3 nM [71] (Table 2). In the latest structural analysis by cryo-EM, the triazopyrimidine group was fitted into the deepest hydrophobic pocket, as in the case of NU7441 and M3814 [39]. Furthermore, the purinone group docked the hydrophobic tunnel formed by Trp3805, Leu 3806 and Met 3829 and was placed mostly parallel with the indole ring of Trp3805, forming π stacking, which might further stabilize the interaction between AZD7648 and DNA-PKcs [39]. 

AZD7648 sensitized cells to IR and doxorubicin at submicromolar concentrations [71] (Table 3). Oral administration of AZD7648 at 37.5–100 mg/kg potentiated tumor growth inhibition by radiation, doxorubicin and olaparib, PARP inhibitor, in vivo [71,117,118,119] (Table 3). It is notable that AZD7648 enhanced the olaparib sensitivity of ATM-deficient cells [71]. This is in contrast to earlier studies showing that the absence or inhibition of DNA-PK, using NU7441 or KU-0060648, alleviated the toxicity of olaparib on ATM-deficient cells [145,146].

Phase 1 and 2 clinical trials of AZD7648 monotherapy and its combination with doxorubicin or olaparib are currently in progress.

## 5. Summary and Future Directions

As seen above, a number of DNA-PK inhibitors have been developed. Potent and selective inhibitors in the earlier generation, such as NU7026 and NU7441, have been useful tools in functional studies of DNA-PK. In the next generation, dual inhibitors such as LY3023414 and CC-115, and DNA-PK-selective inhibitors such as VX-984, M3814 and AZD7648, have promising effects in preclinical studies and are now under clinical trials. The development of these inhibitors in the 1990s and 2000s was greatly promoted by the systematic chemical modification of previously identified PI3K inhibitors to increase the potency and selectivity toward DNA-PK. On the other hand, M3814 and AZD7648 were found recently through large-scale de novo screening, underscoring the importance of this approach. The latest study elucidating the structure of the complex of DNA-PKcs and inhibitors indicated a possibility of structure-guided drug development.

Dual inhibitors LY3023414 and CC-115 have exhibited growth inhibitory effects in monotherapy as seen in preclinical studies. On the other hand, VX-984, M3814 and AZD7648 showed modest growth inhibition at most but potent sensitization to IR and DNA-damaging chemotherapeutic agents. Thus, the growth inhibition by dual inhibitor may be due primarily to the inhibition of PI3K/mTOR. In order to utilize the potential of dual inhibitors in inhibiting DNA-PK, future studies on the combination with radiotherapy and/or chemotherapy are anticipated.

Recent preclinical studies of M3814 and AZD7648 indicate promising combinations. M3814 showed enhancement of the efficacy of chemoradiotherapy and radioimmunotherapy. AZD7648 potentiated the effects of olaparib, especially to a great extent in ATM-deficient cells. There are lines of evidence indicating that the carriers of pathogenic ATM mutation, accounting for 1–2% in human populations, exhibit a several-fold increased risk of breast cancer [147,148]. Since ATM plays a pleiotropic role in the maintenance of the genome, elevated cancer risk may be caused by haploinsufficiency or the second hit, i.e., the loss of the active allele. It is assumed in the latter case that cancer cells have lost ATM function, whereas normal cells retain it. The cancer cells would then be selectively sensitized to the combination of olaparib and DNA-PK inhibitor. Further studies may be warranted to explore the effects of other combinations and in other genetic statuses.

## Figures and Tables

**Figure 1 ijms-23-04264-f001:**
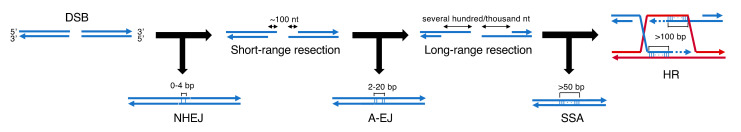
DNA double-strand break pathways. DSB: DNA double-strand break, NHEJ: non-homologous end joining, A-EJ: alternative end joining, SSA: single-strand annealing, HR: homologous recombination.

**Figure 2 ijms-23-04264-f002:**
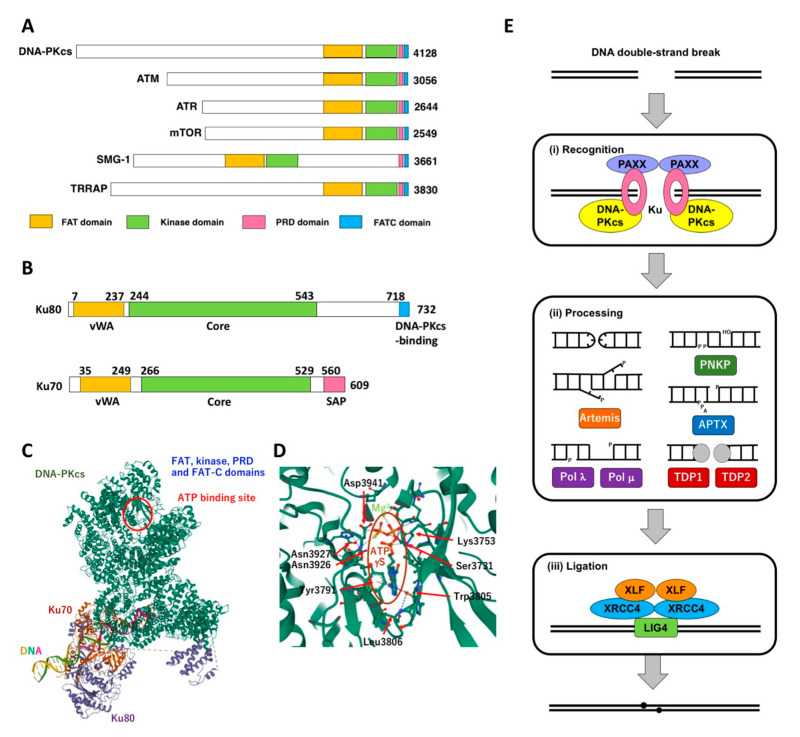
Structure of DNA-PK and its role in NHEJ. (**A**), structure of DNA-PKcs and other PIKK family members. FRAP, ATM and TRRAP (FAT) domain, PIKK-regulatory domain (PRD) and FAT C-terminal (FATC) domains are highlighted. (**B**), structure of Ku80 and Ku70. The von Willebrant factor A (VWA) domain, core domain, SAF-A/B, Acinus and PIAS (SAP) domain and DNA-PKcs binding motif are highlighted. (**C**), the structure of DNA-PKcs complexed with Ku70, Ku80 and dsDNA (RCSB PDB 7K0Y). (**D**), the structure of DNA-PKcs bound to ATPγS (RCSB PDB 7OTP). (**E**), a model for NHEJ. NHEJ proceeds in three stages, (**i**) the recognition stage, (**ii**) the processing stage and (**iii**) the ligation stage. DNA-PK acts in the recognition stage. Figures (**A**,**B**,**E**) are reproduced from our recent reviews [40,59] with some modifications. Figures (**C**,**D**) were drawn using Mol* Viewer [60].

**Figure 3 ijms-23-04264-f003:**
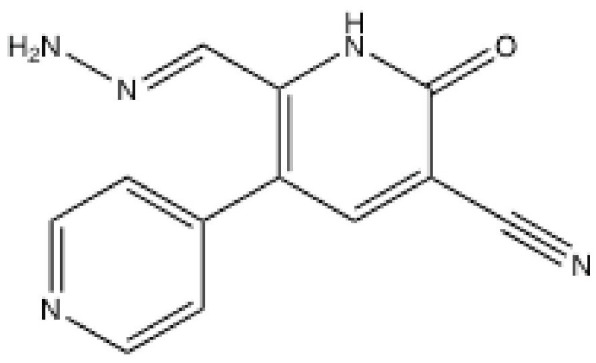
Structure of OK-1035.

**Figure 4 ijms-23-04264-f004:**
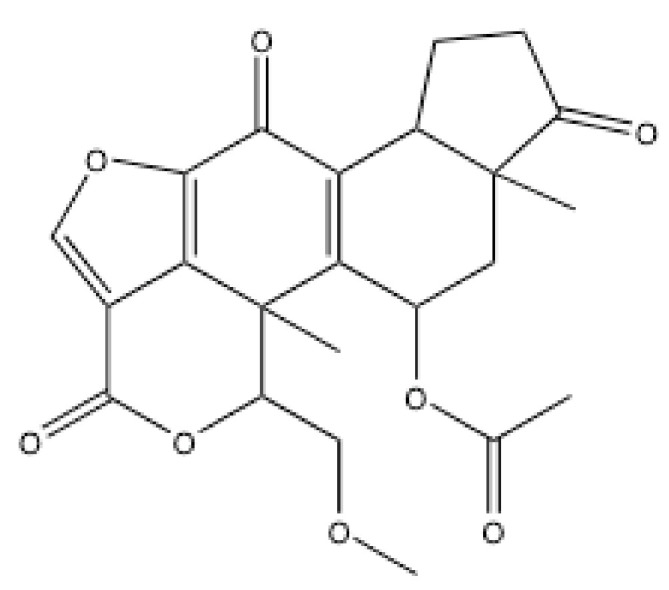
Structure of wortmannin.

**Figure 5 ijms-23-04264-f005:**
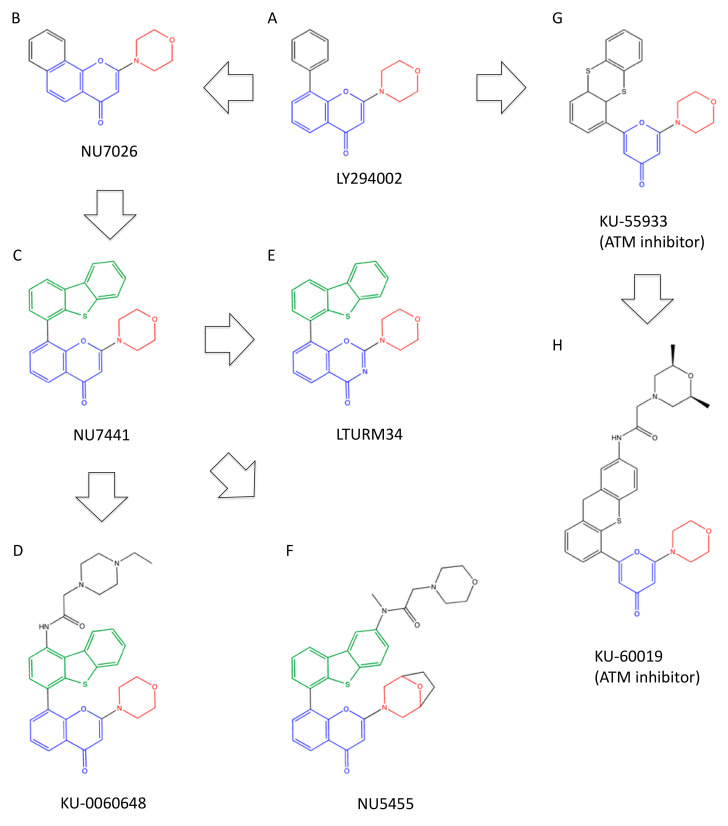
Structures and mutual relationships of LY294002-derived inhibitors. (**A**) LY294002, (**B**) NU7026, (**C**) NU7441, (**D**) KU-0060648, (**E**) LTURM34, (**F**) NU5455, (**G**) KU55933, (**H**) KU-60019. Red: morpholine structures, blue: chromen-4-one or pyran-4-one structures, green: dibenzothiophen structures.

**Figure 6 ijms-23-04264-f006:**
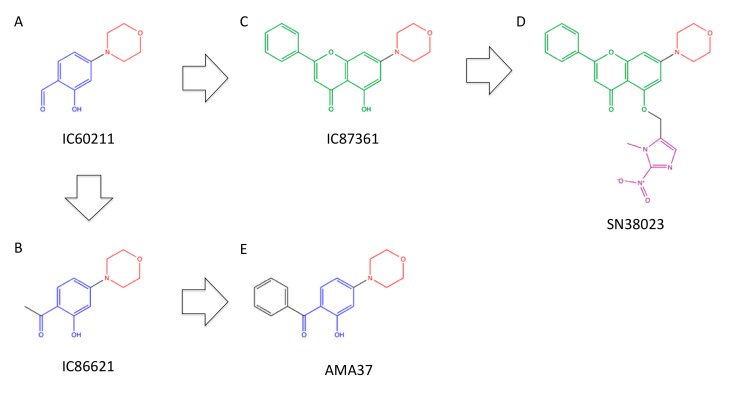
Structures and mutual relationships of arylmorpholine-based inhibitors. (**A**) IC60211, (**B**) IC86621, (**C**) IC87361, (**D**) SN38023, (**E**) AMA37. Red: morpholine structures, blue: o-hydroxybenzaldehyde structures, green: 2-phenyl-4H-chromen-4-one structures, which are common between IC87361 and SN38023, violet: 1-methyl-2-nitro-1H-imidazol-5-yl structure, which is removed in hypoxic conditions.

**Figure 7 ijms-23-04264-f007:**
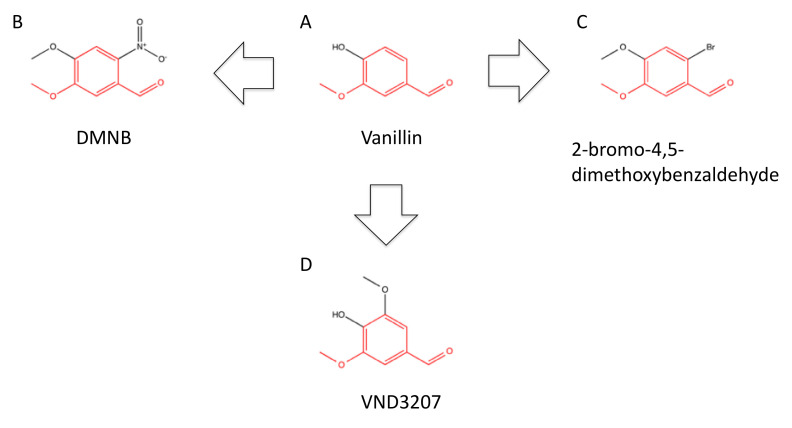
Structure of vanillin-based inhibitors. (**A**) vanillin, (**B**) DMNB, (**C**) 2-bromo-4,5-dimetoxybenzaldehyde, (**D**) VND3207. Meta-methoxy benzaldehyde structures, which are common to all the compounds, are highlighted in red.

**Figure 8 ijms-23-04264-f008:**
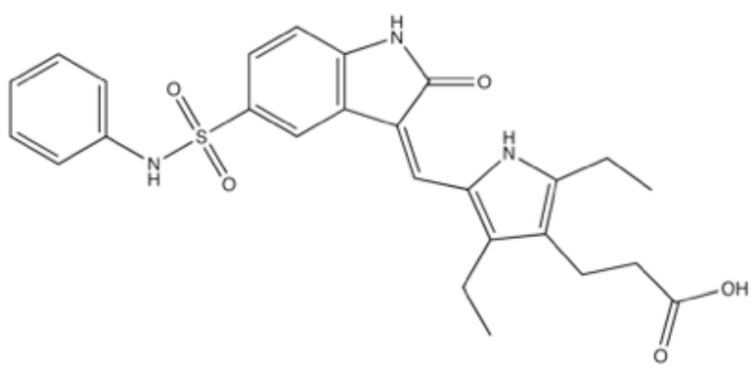
Structure of SU11752.

**Figure 9 ijms-23-04264-f009:**
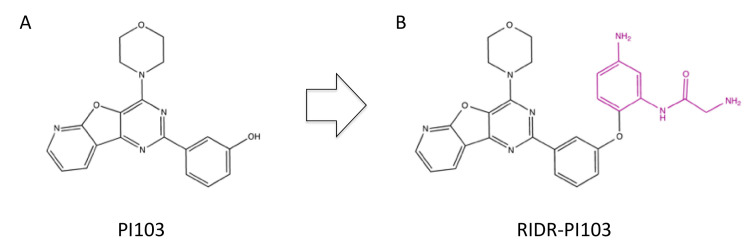
Structure of PI103 (**A**) and RIDR-PI103 (**B**). The phenyl acetamide group, which is removed upon stimulation by reactive oxygen species, is highlighted in violet.

**Figure 10 ijms-23-04264-f010:**
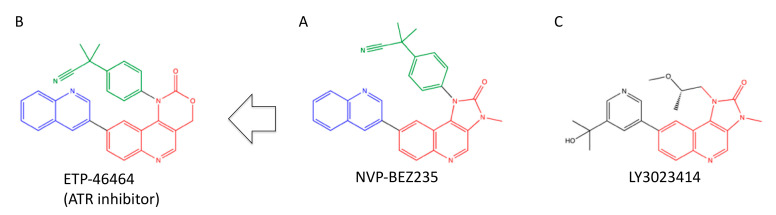
Structures of NVP-BEZ235 (**A**), ETP-46464 (**B**), and LY3023414 (**C**). The imidazoquinoline and oxazinoqinoline structures are highlighted in red. Quinoline and methylphenylpropanenitrile structures, which are common between NVP-BEZ235 and ETP-46464, are highlighted in blue and green, respectively.

**Figure 11 ijms-23-04264-f011:**
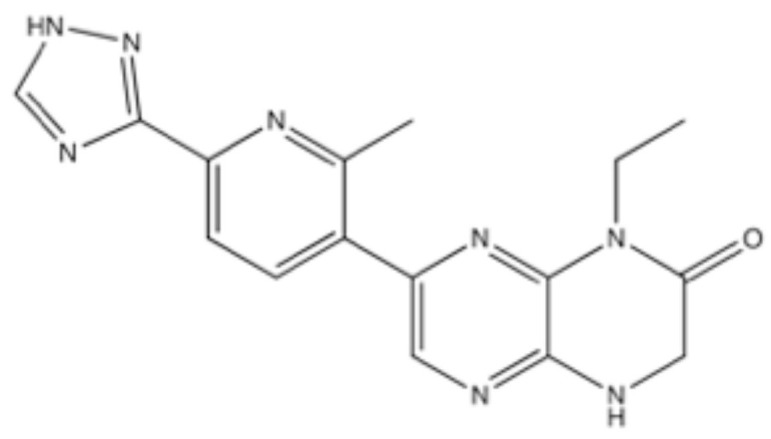
Structure of CC-115.

**Figure 12 ijms-23-04264-f012:**
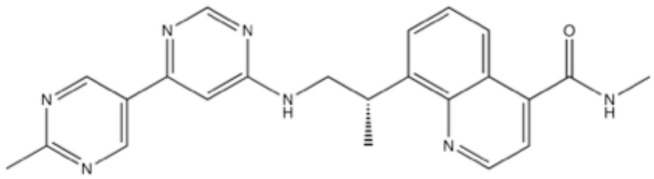
Structure of VX-984.

**Figure 13 ijms-23-04264-f013:**
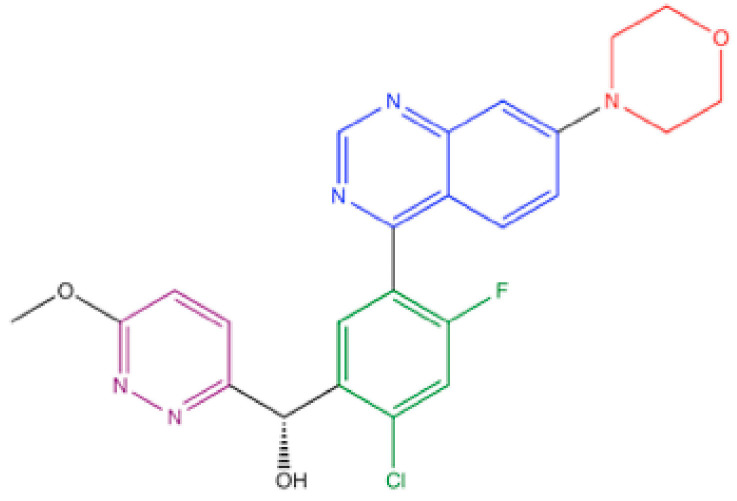
Structure of M3814. Red: morpholine group, blue: quinazoline group, green: chloro-fluorobenzene group, violet: pyridazine group.

**Figure 14 ijms-23-04264-f014:**
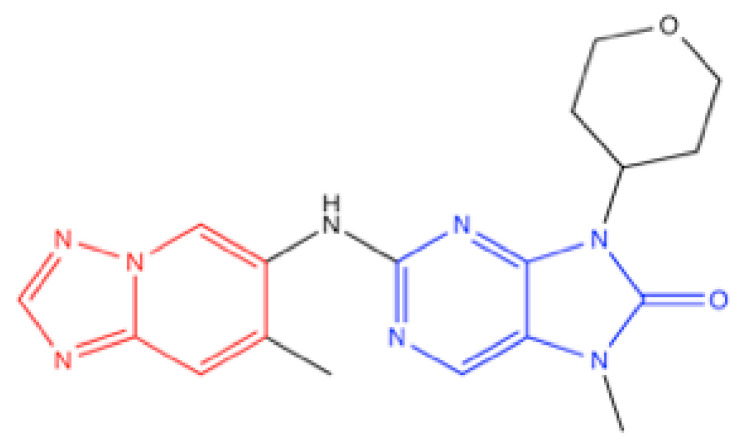
Structure of AZD7648. Red: triazopyrimidine structure, blue: purinone structure.

**Table 1 ijms-23-04264-t001:** DNA-PK substrates and their functions.

Substrates	Function	Substrates	Function
**[DNA Repair and Damage Signaling]**		**[Transcription]**	
**(NHEJ)**		RNA polymerase II	Transcription (general)
DNA-dependent protein kinase catalytic subunit (DNA-PKcs)	DNA-PK complex		
		TATA box-binding protein (TBP)	
Ku autoantigen 80kDa subunit (Ku80)		p53	Transcription (specific)
		Specificity protein 1 (Sp1)	
Ku autoantigen 70 kDa subunit (Ku70)		c-Jun	
		c-Fos	
DNA ligase IV (LIG4)	Ligation complex	c-Myc	
X-ray repair cross-complementing group 4 (XRCC4)		Octamer-binding factor 1 (Oct-1)	
XRCC4-like factor (XLF)		Serum response factor (SRF)	
Artemis	Nuclease	**[RNA metabolism]**	
Polynucleotide kinase phosphatase (PNKP)	Kinase, phosphatase	Nuclear DNA helicase II (NDHII)	Transcription and RNA processing
Werner syndrome protein (WRN)	Helicase, nuclease	Heterogeneous nuclear ribonucleoprotein A1 (hnRNP-A1)	RNA splicing
**(Other DNA repair and damage signaling pathways)**			
Ataxia telangiectasia mutated (ATM)	Protein kinase; HR and cell cycle checkpoint	Heterogeneous nuclear ribonucleoprotein U (hnRNP-U)	
Replication protein A 2 (RPA2)	Single-stranded DNA binding; HR and DNA replication	Fused in sarcoma (FUS)	RNA binding
Poly(ADP-ribose) polymerase 1 (PARP1)	Single-strand break repair	**[Signaling]**	
Excision repair cross complementing 1 (ERCC1)	Nuclease component; nucleotide excision repair	Akt1	Protein kinase
		Akt2	Protein kinase
**[DNA replication]**		Sty1/Spc1-interacting protein 1 (Sin1)	Protein kinase regulator
DNA ligase I (LIG1)	Ligation		
Minichromosome maintenance 3 (MCM3)	Initiation of replication	**[Organelle, cytoskeleton]**	
**[Nucleosome and chromatin structure]**		Golgi phosphoprotein 3 (GOLPH3)	Linking Golgi membrane to cytoskeleton
Histone H2AX	Core histone component; recruitment of DSB repair proteins	Vimentin	Intermediate filament
Histone H1	Linker histone	Tau	Microtubule regulation
High mobility group 1 (HMG1)	Maintenance and regulation of chromatin structure	**[Protein maintenance]**	
High mobility group 2 (HMG2)		Heat shock protein 90 alpha (HSP90a)	Protein chaperone
C1D		Valosin-containing protein (VCP)	AAA+ ATPase
Topoisomerase I	Regulation of topological status of DNA	**[Metabolism]**	
Topoisomerase II		Fumarate hydratase (FH)	Production of L-malate from fumarate; regulation of NHEJ
Nuclear orphan receptor 4A2 (NR4A2)	Chromatin regulation; regulation of NHEJ		
Pituitary tumor-transforming gene (PTTG)	Regulation of chromosome segregation		

**Table 2 ijms-23-04264-t002:** DNA-PK inhibitors with IC_50_ for PIKKs and PI3Ks.

Name of Inhibitor	IC_50_ (nM)								Ref.
	DNA-PK	ATM	ATR	mTOR	PI3Kα	PI3Kβ	PI3Kγ	PI3Kδ	
OK-1035	8000								[61]
	100,000								[62]
Wortmannin	16	150							[65]
	120								[66]
	260	300	4400	2500	3				[67]
LY294002	6000								[67]
	1400	>10,000	>10,000	2800	300	270	3020	220	[68]
NU7026	230	>100,000	>100,000	6400	13,000				[69]
NU7441	14	>100,000	>100,000	1700	5000				[70]
	40	>10,000	>10,000	2400	130	16	220	30	[68]
	185	>3100	>30,000	1800	7800				[71]
KU-0060648	5	>10,000	>10,000	10,000	4	0.5	590	<0.1	[68]
	55	>30,000	>30,000	150	200				[71]
LTURM34	34				>10,000	5,800	>10,000	8500	[72]
NU5455	8.2	>10,000	>10,000	4058	1870	9320	>10,000	276	[73]
IC60211	400								[74]
IC86621	120				1400	135	880	1000	[74]
IC87361	34				3800	1700	800	7900	[74]
AMA37	270	>100,000	>100,000	>100,000	32,000	3700	~100,000	22,000	[75]
Vanillin	1,500,000								[76]
DMNB	15,000								[76]
2-bromo-4,5-dimethoxybenzaldehyde	30,000								[76]
SU11752	130						1100		[77]
PI103	14				2	3	15	3	[78]
	7.5				8	7	15	172	[79]
NVP-BEZ235	1.7				7	72	6	38	[79]
	5	7	21	2	2				[80]
LY3023414	4.24			165	6.07	77.6	23.8	38	[81]
CC-115	13			21	852				[82]
VX-984	115	>30,000	>30,000	>30,000	>20,000	>30,000	7100	>30,000	[71]
M3814	0.6–20	10,000	2800	>10,000	330	250	>1000	95	[83]
	43	>30,000	>30,000	550	800	170	1590	350	[71]
AZD7648 ^1^	91.3	17,930	>29,770	>30,000	>8030	>30,000	1370	>30,000	[71]

^1^ IC_50_ values for in cellulo phosphorylation of target proteins are shown.

**Table 3 ijms-23-04264-t003:** In cellulo and in vivo radio- or chemo-sensitizing effects of DNA-PK inhibitors.

Name of Inhibitor	In Cellulo Sensitizing Effect					In Vivo Sensitizing Effect				
	Radiation		Chemotherapeutic Drug			Radiation		Chemotherapeutic Drug		
	μM	Ref.	μM	Drug	Ref.	mg/kg ^1^	Ref.	mg/kg ^1^	Drug	Ref.
NU7026	10	[69]	10	idarubicin, daunorubicin, doxorubicin, etoposide, amsacrine (mAMSA), mitroxantrone	[89]	25, i.p.	[91]	50, i.p.	salinomycin	[92]
								50, i.p.	TIC10 ^3^	[93]
NU7441	0.5	[95]	0.5	etoposide	[95]	25, i.p.	[96]	10, i.p.	etoposide	[95]
KU-0060648	0.1	[68]	1	etoposide, doxorubicin	[97]			10, i.p.	etoposide	[97]
			1-10	temozolomide	[99]			10, 50, i.p.	temozolomide	[98]
LTURM34			3	docetaxel	[100]					
NU5455	1	[73]	1	etoposide, doxorubicin	[73]	30, p.o.	[73]	100, p.o.	etoposide	[73]
								30, p.o.	doxorubicin	[73]
IC86621	50	[74]				400, s.c.	[74]			
IC87361	7	[74]				75 ^2^, i.p.	[74]			
AMA37	20	[103]								
Vanillin	100, 300	[76]	100	cisplatin	[76]					
DMNB			15	cisplatin	[76]					
SU11752	50	[77]								
PI103	0.06–1	[104]	0.06–1	doxorubicin, etoposide, temozolomide	[104]					
NVP-BEZ235	0.1	[105]				50, 75, p.o.	[106]			
LY3023414								15, p.o.	rapamicin, cisplatin+gemcitabin	[81]
CC-115	1	[107]								
VX-984	0.1–0.5	[108]				50, p.o.	[108]			
M3814	1	[83]	0.3–0.9	calichiamicin	[109]	5–50, p.o.	[110]	100, p.o.	Mylotarg	[110]
	0.111–1	[110]	0.3	daunorubicin	[111]	50, p.o.	[112]	50, i.g.	paclitaxel, etoposide	[113]
	0.5–15	[112]	5	paclitaxel, etoposide	[113]			50, p.o.	PLD ^4^	[114]
						50, p.o.	IR + 5-FU			[115]
						50, p.o.	IR + bintrafusp alpha			[116]
AZD7648	0.1, 1	[71,117,118]	0.1	doxorubicin	[71]	50, 100, p.o.	[71]	37.5, 75, p.o.	doxorubicin, olaparib	[71]
						75, p.o.	[117,118]	100, p.o.	PLD, olaparib	[119]

^1^ Abbreviations for the route of administration: i.p., intraperitoneal injection; i.g., intragastrical injection; p.o., per os (oral administration); s.c., subcutaneous injection. ^2^ Unit: μg/animal. ^3^ TIC10: TRAIL-inducing compound 10. ^4^ pegylated liposomal daunorubicin.

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
