# Peer review of "Development and Evolution of DNA-Dependent Protein Kinase Inhibitors toward Cancer Therapy"

_ijms, 2022, doi:10.3390/ijms23084264_

Round 1

Reviewer 1 Report

This review by Matsumoto details the actions of a substantial number of compounds that have been identified to inhibit the DNA-dependent protein kinase (DNA-PK) and its PI3K homologs. While the review contains much useful information on the structures, potency and selectivity of individual inhibitors, it could be improved substantially by inclusion of additional information and clarification of the writing. Since many of the inhibitors also suppress the action of PI3K kinases other than DNA-PK, the author could have included additional information on these other kinases and their roles in the DNA damage response and cancer. Comments for improvement are below.

Specific comments

  1. The review contains limited information about the PI3K kinases other than DNA-PK, even though inhibition of other PI3K kinases is mentioned throughout the manuscript. This is particularly problematic considering the potential efficacy of the dual inhibitors that target DNA-PK and mTOR kinase. The review would be strengthened by a more in-depth treatment of the functions of other PI3K kinases and their roles in DNA damage responses and cancer development and progression.
  2. The author could also include information or at least mention the alt-NHEJ pathway that seems to serve as a back-up for the canonical NHEJ and HR pathways for DSB repair. In this regard, the author should present a clearer picture as to the role of canonical NHEJ and specifically DNA-PK. (lines 42-43) This statement implies that 80% of breaks occurring in G2 phase are repaired via NHEJ, which is not cited and seems to be in direct contrast with the statement on lines 38-39. (lines 87-94) Description of processing enzymes and their activities in the context of the overall repair reactions catalyzed by NHEJ is confusing as written. The lack of clarity on the canonical NHEJ pathway, DNA-PK and associated repair factors undermines the remainder of the manuscript.
  3. Since the review focuses on DNA-PK inhibition, it is necessary to at least provide a table of protein factors phosphorylated by DNA-PK if not other PI3K homologs. A list of DNA-PK targets would provide readers with valuable information about what inhibitors might do in the cellular setting.
  4. The writing of the manuscript is poor in many places, and its clarity suffers as a result. The manuscript should be carefully proofread throughout for grammar and proper English.

Minor comments

  1. (lines 28-30) This statement actually underestimates the burden of DNA damage caused by ionizing radiation since purine base damage has not been mentioned here. Double-strand breaks are the most deleterious, not necessarily the most “critical.”
  2. (Lines 124-125) Author should clarify the IC50 of OK-1035 if possible.
  3. (line 196-201) Clarify whether the compound in question is NU5455 or NU7455.
  4. (lines 285-287) Sentence about ETP-46464 is out of place and should be moved to the end of this section, as the remainder of the material focuses on NVP-BEZ235.
  5. Tables 1 and 2 are perhaps the most useful information provided in the manuscript, but I could not find that they were referred to at all in the text. (Table 2) Correct “sensitizing’ in both column titles.

Author Response

Thank you very much for careful reading of my manuscript and valuable comments.

Specific comments

  1. The review contains limited information about the PI3K kinases other than DNA-PK, even though inhibition of other PI3K kinases is mentioned throughout the manuscript. This is particularly problematic considering the potential efficacy of the dual inhibitors that target DNA-PK and mTOR kinase. The review would be strengthened by a more in-depth treatment of the functions of other PI3K kinases and their roles in DNA damage responses and cancer development and progression.

I agree your opinion. It is important to note that PI3K and mTOR are implicated in radiosensitivity. I added this information in the Appendix.

  1. The author could also include information or at least mention the alt-NHEJ pathway that seems to serve as a back-up for the canonical NHEJ and HR pathways for DSB repair. In this regard, the author should present a clearer picture as to the role of canonical NHEJ and specifically DNA-PK. (lines 42-43) This statement implies that 80% of breaks occurring in G2 phase are repaired via NHEJ, which is not cited and seems to be in direct contrast with the statement on lines 38-39. (lines 87-94) Description of processing enzymes and their activities in the context of the overall repair reactions catalyzed by NHEJ is confusing as written. The lack of clarity on the canonical NHEJ pathway, DNA-PK and associated repair factors undermines the remainder of the manuscript.

First, I reinforced the description of DSB repair, including alternative end joining and single-strand annealing (page 1, line 33-page 2, line 63; page 2, line 74-76). I also added a new Figure 1.

Second, I added the reference [2] for the statement that “even in G2 phase, approximately 80% of DSBs are thought to be repaired via NHEJ in human cells”. It is not contradictory to the statement that “HR is restricted to late S and G2 phases”. It means that HR has smaller contribution to NHEJ even in G2 phase.

Third, I rewrote the description of NHEJ, dividing into three stages according to the Figure 2, E (page 3, line 287-301). The description of the processing enzymes has been edited by two native English speakers.

  1. Since the review focuses on DNA-PK inhibition, it is necessary to at least provide a table of protein factors phosphorylated by DNA-PK if not other PI3K homologs. A list of DNA-PK targets would provide readers with valuable information about what inhibitors might do in the cellular setting.

I added a table of list for DNA-PK substrates (Table 1).

  1. The writing of the manuscript is poor in many places, and its clarity suffers as a result. The manuscript should be carefully proofread throughout for grammar and proper English.

The manuscript has been checked and corrected by two native English speakers (Anie Day DC Asa and Amanda Mando Eijansantos) in my laboratory.

Minor comments

  1. (lines 28-30) This statement actually underestimates the burden of DNA damage caused by ionizing radiation since purine base damage has not been mentioned here. Double-strand breaks are the most deleterious, not necessarily the most “critical.”

“Critical” has been replaced by “deleterious” (page 1, abstract line 8; line 30).

  1. (Lines 124-125) Author should clarify the IC50 of OK-1035 if possible.

The reason for discrepancy in IC50 of OK-1035 between two reports is unclear.

  1. (line 196-201) Clarify whether the compound in question is NU5455 or NU7455.

I apologize for this mistake, incurring confusion. NU5455 is correct (page 8, lines 783 and 786).

  1. (lines 285-287) Sentence about ETP-46464 is out of place and should be moved to the end of this section, as the remainder of the material focuses on NVP- BEZ235.

It was moved to the end of the section (page 12, line 935-937).

  1. Tables 1 and 2 are perhaps the most useful information provided in the manuscript, but I could not find that they were referred to at all in the text. (Table 2) Correct “sensitizing’ in both column titles.

The tables are referred to in the text. I also corrected the spelling mistake.

Reviewer 2 Report

Dear Yoshihisa Matsumoto,

thank you very much for submitting your review on the DNA-PK inhibitors to the International Journal of Molecular Science.

Here are my comments:

The review provides an extensive summary of the different DNAPKcs inhibitors. It starts with a short introduction of the biological function of the kinase before summarising the different inhibitors further divided in non-clinical and clinical kinase blockers.

L15  I not `we` as there is only one author

L34 the point of the NHEJ is not only close vicinity but importantly the independency of DNA sequence homology at either DNA end – please make this clear here

L46 please ad sentence or two regarding the importat role of NHEJ in antibody diversity (ie the homology-independent function of DNAPK)

L53 please add the word “kinase” after (ATR)

L94 Adenosinemonophosphate (AMP)

I would like to suggest to add one short additional section after Figure 1 in which you describe the structure of DNPKcs (ie. Chen X, Xu X, Chen Y, Cheung JC, Wang H, Jiang J, de Val N, Fox T, Gellert M, Yang W. Structure of an activated DNA-PK and its implications for NHEJ. Mol Cell. 2021 Feb 18;81(4):801-810.e3. doi: 10.1016/j.molcel.2020.12.015. Epub 2020 Dec 31. PMID: 33385326) as this is very important to understand the ability of the molecules to block the kinase. Please emphasise the binding site(s) of the inhibitors on the kinase structure (ie ATP binding pocket (Fig 2 in Ref 53)

L115 related to the above comment, please describe briefly the key structural similarities between the three kinases that formed the basis for the development of the inhibitors

In section 3.3, please make it clearer that the inhibitors NU7441, KU-0060648 and LTURM34 are further developments of LY294002 as shown in Fig 4. Please make a reference to Fig 4

Please summarise at the beginning of section 4 the key differences between the non-clinical and clinical inhibitors.

Author Response

Thank you very much for carefully reading my manuscript and valuable comments.

L15 I not `we` as there is only one author

Thank you. It was corrected (page 1, abstract, line 15).

L34 the point of the NHEJ is not only close vicinity but importantly the independency of DNA sequence homology at either DNA end – please make this clear here

I agree your opinion. Now I referred first to the requirement for little or no sequence homology and then to place of spatial vicinity (page 1, line 39; page 2, line 66).

L46 please ad sentence or two regarding the important role of NHEJ in antibody diversity (ie the homology- independent function of DNA-PK)

I added a brief explanation of V(D)J recombination at the end of Introduction (page 2, line 77-83).

L53 please add the word “kinase” after (ATR)

It was corrected accordingly (page 2, line 95).

L94 Adenosinemonophosphate (AMP)

It was corrected accordingly (page 3, line 296).

I would like to suggest to add one short additional section after Figure 1 in which you describe the structure of DNA-PKcs (ie. Chen X, Xu X, Chen Y, Cheung JC, Wang H, Jiang J, de Val N, Fox T, Gellert M, Yang W. Structure of an activated DNA-PK and its implications for NHEJ. Mol Cell. 2021 Feb 18;81(4):801-810.e3. doi: 10.1016/j.molcel.2020.12.015. Epub 2020 Dec 31. PMID: 33385326) as this is very important to understand the ability of the molecules to block the kinase. Please emphasise the binding site(s) of the inhibitors on the kinase structure (ie ATP binding pocket (Fig 2 in Ref 53)

I added a paragraph about the structure of DNA-PKcs complexed with Ku, DNA and ATP analog, citing these literatures (page 3, line 255-272). The snapshots are also added in Figure 2 (Figure 1 in original version).

L115 related to the above comment, please describe briefly the key structural similarities between the three kinases that formed the basis for the development of the inhibitors

In the added paragraph, I noted the conserved and non-conserved residues among PIKKs, which are in contact with ATP (page 3, line 264-272).

In section 3.3, please make it clearer that the inhibitors NU7441, KU-0060648 and LTURM34 are further developments of LY294002 as shown in Fig 4. Please make a reference to Fig 4

To make it clearer that the inhibitors such as NU7441 are derived from LY294002, I added the phrase “such as NU7026 and NU7441” (page 7, line 691) and reference to Figure 5 (Figure 4 in the original version), following “more potent and selective inhibitors of DNA-PK”. In the description of each inhibitor, Figure 5 had been already referred to.

Please summarise at the beginning of section 4 the key differences between the non-clinical and clinical inhibitors.

I had already mentioned that pharmacokinetics and toxicity are key issues for the clinical use. I added the issue of solubility for oral availability and also increased contacts from the structural aspect (page 12, line 913-918).

Round 2

Reviewer 1 Report

The revision of this review by Matsumoto is much improved. Below are some minor issues that could be addressed in the final version.

Minor points:

  1. (line 111) If the Ser3731, Leu3806 and Asn3926 residues are not conserved, then they would be divergent or not conserved instead of “not divergent.”
  2. (lines 123-126) I think most DNA repair researchers would consider Artemis as a core NHEJ component.
  3. Table 1 needs to be reorganized and clarified, perhaps with a focus on factors that participate in repair reactions/DNA damage response pathways.
  4. It might be better to integrate information on SMG-1 and mTOR kinases into section 2 instead of including in an Appendix at the end of the manuscript.
  5. Although the writing is much improved, the manuscript still contains a number of minor grammatical issues.

Author Response

The revision of this review by Matsumoto is much improved. Below are some minor issues that could be addressed in the final version.

Thank you very much for valuable comments again.

Minor points:

1. (line 111) If the Ser3731, Leu3806 and Asn3926 residues are not conserved, then they would be divergent or not conserved instead of “not divergent.”

I apologize for this mistake. It is corrected.

2. (lines 123-126) I think most DNA repair researchers would consider Artemis as a core NHEJ component.

I agree that Artemis is often considered as a core NHEJ component. I have rearranged the description of NHEJ process.

3. Table 1 needs to be reorganized and clarified, perhaps with a focus on factors that participate in repair reactions/DNA damage response pathways.

I have reorganized Table 1 so that the proteins involved in DNA repair reactions and DNA damage response pathways are listed first.

4. It might be better to integrate information on SMG-1 and mTOR kinases into section 2 instead of including in an Appendix at the end of the manuscript.

According to this suggestion, the materials referring to SMG-1 and PI3K/mTOR are moved to section 2.

5. Although the writing is much improved, the manuscript still contains a number of minor grammatical issues.

The entire manuscript was carefully read again by two native English speakers. I have made corrections in spelling and grammar as marked up by yellow. I have also reorganized several sentences for clarification as marked up by green.